# Non-enzymatic primer extension with strand displacement

**Lijun Zhou[1,2,3], Seohyun Chris Kim[1,2,3], Katherine H Ho[4], Derek K O'Flaherty[1,2,3†], Constantin Giurgiu[4], Tom H Wright[1,2,3], Jack W Szostak[1,2,3,4]\***

[1]Department of Molecular Biology, Howard Hughes Medical Institute, Massachusetts General Hospital, Boston, United States; [2]Center for Computational and IntegrativeBiology, Massachusetts General Hospital, Boston, United States; [3]Department of Genetics, Harvard Medical School, Boston, United States; [4]Department of Chemistry and Chemical Biology, Harvard University, Cambridge, United States

**Abstract** Non-enzymatic RNA self-replication is integral to the emergence of the 'RNA World'. Despite considerable progress in non-enzymatic template copying, demonstrating a full replication cycle remains challenging due to the difficulty of separating the strands of the product duplex. Here, we report a prebiotically plausible approach to strand displacement synthesis in which short 'invader' oligonucleotides unwind an RNA duplex through a toehold/branch migration mechanism, allowing non-enzymatic primer extension on a template that was previously occupied by its complementary strand. Kinetic studies of single-step reactions suggest that following invader binding, branch migration results in a 2:3 partition of the template between open and closed states. Finally, we demonstrate continued primer extension with strand displacement by employing activated 3'-aminonucleotides, a more reactive proxy for ribonucleotides. Our study suggests that complete cycles of non-enzymatic replication of the primordial genetic material may have been facilitated by short RNA oligonucleotides.

**\*For correspondence:**
szostak@molbio.mgh.harvard.edu

**Present address:** †Alnylam Pharmaceuticals, Cambridge, United States

**Competing interests:** The authors declare that no competing interests exist.

## Introduction

The replication of a genetic polymer within a vesicle capable of growth and division may allow a protocell to undergo Darwinian evolution. The construction of a protocell capable of autonomous reproduction, even within an artificial laboratory setting, may lead to significant insights into the origin of life (*Joyce and Szostak, 2018*). Prior to the emergence of enzymes or ribozymes on the early Earth, the first genetic material had to rely on non-enzymatic copying reactions to accomplish self-replication. Following template-directed RNA copying, the daughter strand, which is complementary to the template, must undergo another round of copying to generate a product with the same sequence as the original template, thus completing the replication process. Recent discoveries have greatly improved the rate and extent of chemical RNA copying reactions in laboratory studies (*Li et al., 2017*; *O'Flaherty et al., 2018*). However, the important question of how to non-enzymatically complete a true replication cycle remains challenging. After the first round of templated copying, the daughter strand is sequestered within a stable duplex with its parent strand. Even if heat is used to temporarily separate the two strands, when cooled down, the rate of reannealing of the two strands is much faster than the rate at which template copying occurs (*Szostak, 2012*). This 'strand reannealing' or 'strand inhibition' effect is a major obstacle to non-enzymatic genetic replication under prebiotically plausible conditions.

Due to the importance of the strand separation problem, several approaches have been explored in efforts to enable cycles of replication. The Hud group has employed highly viscous solvent mixtures (glycholine) to slow the reannealing of complementary strands, together with temperature

cycling to repeatedly separate the strands of RNA duplexes (*He et al., 2017*). Using this approach they were able to demonstrate that 11 individual 32-mer RNAs could be enzymatically ligated on a 545 bp RNA template in the presence of the full length complementary strand. However, this method may not be compatible with the much slower non-enzymatic RNA ligation or polymerization, and may also not work as well for the copying of shorter templates that will still diffuse rapidly through viscous solvents. The Richert group reported an example of genetic polymer replication, using bead immobilization and iterative cycles of protection/deprotection (*Hänle and Richert, 2018*). While providing a powerful method to study the chemical replication of genetic polymers in a laboratory context, this approach is not prebiotically relevant and is also incompatible with a proto-cellular context. The Sutherland group has reported that pH fluctuations can drive RNA separation (*Mariani et al., 2018a*), but this does not address the problem of rapid strand annealing following the return of the pH to more neutral values. The Braun group has recently shown that wet-dry cycles inside heated rock pores triggering salt concentration fluctuations can also lead to strand separation (*Ianeselli et al., 2019*). Although these latter two approaches are simple and prebiotically plausible, up till now no demonstration of non-enzymatic RNA copying has been reported with these methods. Here, we explore a prebiotically plausible solution to the strand separation problem that enables non-enzymatic copying reactions to proceed in the presence of a complementary template-bound strand, and that is in principle compatible with operation in a protocellular context.

In extant biology, genomic replication never occurs by strand separation followed by template copying; instead, duplex unwinding and primer extension occur in concert via strand displacement synthesis (*Benkovic et al., 2001*). This approach has not typically been considered relevant to prebiotic replication because of the requirement for highly sophisticated enzyme catalysts. However, the concept of strand displacement by branch migration, originally developed in the field of genetic recombination (*Holliday, 1964*) has been widely used in the fields of DNA and RNA nanotechnology (*Zhang and Seelig, 2011*) and the study of RNA function (*Bhadra and Ellington, 2014*). Inspired by the precedent from biology and the powerful methods of nucleic acid nanotechnology, we asked whether we could use the binding of short oligonucleotides to unwind an RNA duplex and trigger non-enzymatic primer extension reactions.

In the non-enzymatic system we investigate, primer extension with activated nucleotides occurs via reaction of the primer 3'-hydroxyl with a 5'−5'-phosphorimidazolium-bridged dinucleotide, which is an obligate covalent intermediate in non-enzymatic primer extension that is formed by the reaction of two 2-aminoimidazole activated ribonucleotides with each other (*Figure 1A*) (*Walton and Szostak, 2017*; *Zhang et al., 2018*). The 2-aminoimidazole activated ribonucleotides are in turn generated by the reaction of nucleotides with 2-aminoimidazole in a process driven by methyl isocyanide (*Mariani et al., 2018b*), and 2-aminoimidazole itself can be synthesized under prebiotically reasonable conditions (*Fahrenbach et al., 2017*).

Here, we report that using short oligonucleotides as 'invaders' we can copy an RNA templating region that is already occupied by its complementary strand. We first explored this concept in the context of the addition of a single nucleotide to a primer through reaction with an imidazolium-bridged dinucleotide. We then iterated this process to demonstrate primer extension by multiple nucleotides, using multiple invaders. This study demonstrates a potential solution to the strand inhibition problem and thus represents a step towards the realization of non-enzymatic RNA replication.

## Results

To test whether non-enzymatic RNA synthesis could proceed via spontaneous strand displacement synthesis, we prepared RNA primer/template complexes in which the templating region was either open (*Figure 1B*) or occupied by a complementary strand (the 'blocker') with a 6-nucleotide (nt) toehold region at its 5'-end (*Figure 1C*). As the substrate for primer extension, we used the 5'−5'-imidazolium-bridged diribocytosine (C*C) intermediate for copying the GG template. With an open template, the C*C intermediate base pairs with the GG template, followed by attack of the 3'-hydroxyl group of the RNA primer on the adjacent phosphate of the C*C, resulting in rapid +1 extension of the primer and release of one free *C as the leaving group (*Figure 1E*). In the presence of the template blocking strand, defined as a closed conformation, the C*C cannot base pair with the template, and no primer extension was observed (*Figure 1F*). Thus, as expected, non-enzymatic primer extension cannot occur by strand displacement synthesis without an auxiliary.

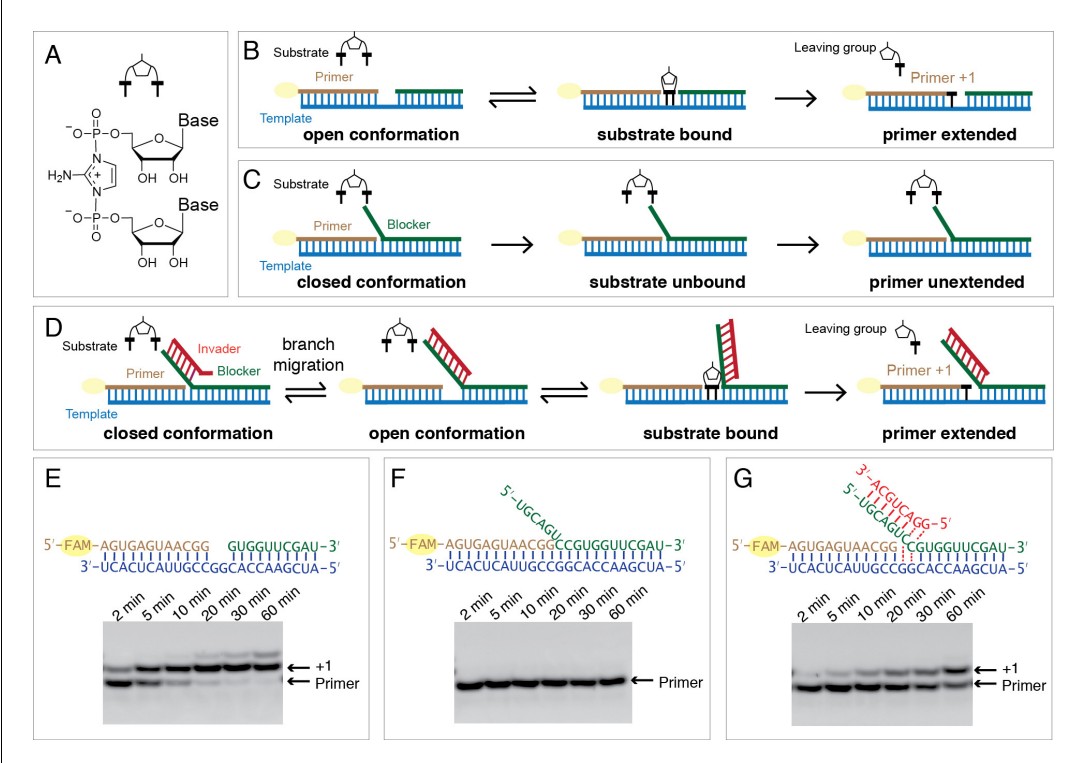

**Figure 1.** Single step non-enzymatic primer extension with strand displacement. (**A**) Chemical structure of the 5′−5′-phosphorimidazolium-bridged dinucleotide. (**B**)-(**D**) Experimental design illustrations in three different cases. (**E**) PAGE analysis of a primer extension reaction corresponding to (**B**), the imidazolium-bridged dinucleotide C*C binds the open GG templating region and reacts with the 3′-end of the primer, resulting in +1 extension product. (**F**) PAGE analysis of a primer extension reaction corresponding to (**C**). The blocker (green), an oligonucleotide complementary to the templating region, inhibits the primer extension reaction. (**G**) PAGE analysis of a primer extension reaction corresponding to (**D**). An invader (red), a short oligonucleotide partially complementary to the blocker, is able to rescue the primer extension reaction by sequestering the blocker and allowing C*C to bind and react. All primer extension reactions were conducted at room temperature, 50 mM Na⁺-HEPES, pH 8.0, 50 mM MgCl₂, 3 mM C*C, 1.5 μM primer, 2.5 μM template, 0 or 3.5 μM blocker, 0 or 5 μM invader.

We then tested the hypothesis that a short 'invader' RNA strand could open up the template region next to the primer by first binding to the 5′ overhang, or toehold region, of the blocker strand, followed by branch migration (*Figure 1D*). Full base-pairing of the invader and blocker strands should release the template and allow it to bind the C*C substrate, defined as an open conformation. As an initial test, we employed an 8-nt RNA oligonucleotide as an invader, with a 6-nt region complementary to the 6-nt toehold of the blocker, plus a 5′-GG sequence to allow for extended base-pairing with the blocker. If the invader was able to fully pair with the blocker by competing for template-blocker pairing, the template should be freed, allowing C*C binding to occur. Strikingly, after C*C addition to the primer/template/blocker complex in the presence of the invader, we observed the appearance of the +1 product of primer extension (*Figure 1G*). The invader-mediated primer extension reaction was slower than primer extension on an open template, presumably reflecting the equilibrium between the open and closed template states in the presence of the invader oligonucleotide. This experiment demonstrates that primer extension by strand displacement synthesis can be facilitated by short RNA oligonucleotides.

Encouraged by our initial observation of non-enzymatic strand displacement synthesis, we sought to optimize this process by varying invader length and concentration. We examined primer extension reactions with 6-nt and 8-nt invaders as a function of concentration, and calculated the pseudo-first order reaction rate, $k_{obs}$, from the disappearance of unreacted primer versus time (*Figure 2*). For the octamer invader at room temperature, $k_{obs}$ increased as a function of invader concentration until a maximal rate of $0.9 \pm 0.1$ $h^{-1}$ was reached at a concentration of 5 μM. For the hexamer invader, essentially no primer extension could be observed when the concentration was 15 μM or

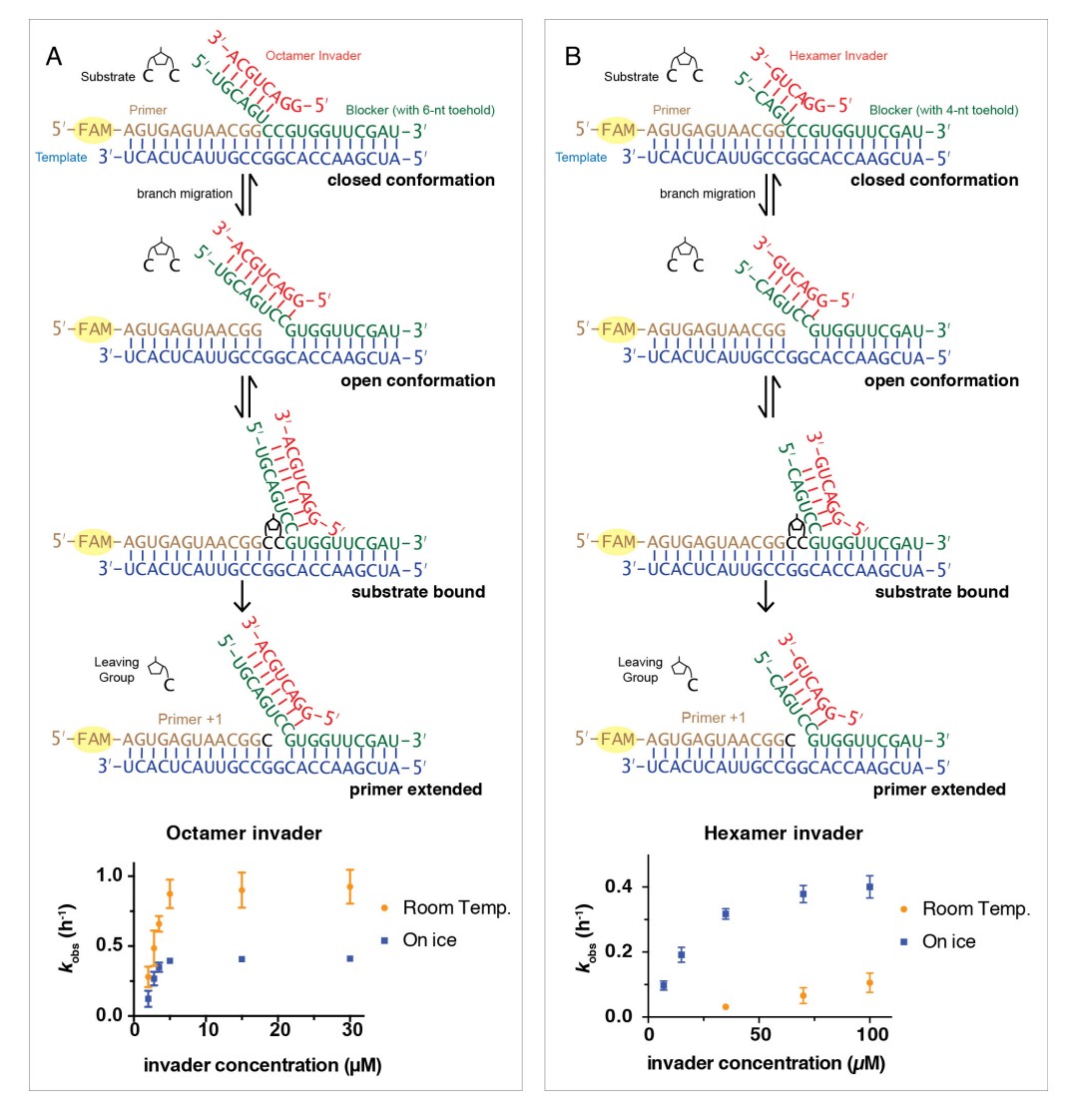

**Figure 2.** Rates of primer extension reactions as a function of invader concentration. (**A**) Reactions with the blocker possessing a 6-nt toehold, and an 8-nt long invader. (**B**) Reactions with the blocker possessing a 4-nt toehold, and a 6-nt long invader. All primer extension reactions were conducted in 50 mM Na$^+$-HEPES, pH 8.0, 50 mM MgCl$_2$, 3 mM C*C, 1.5 µM primer, 2.5 µM template, 3.5 µM blocker, at room temperature or on ice as indicated. Reaction products were analyzed by denaturing PAGE. (n ≥ 3, The error bars are smaller than the symbol when they are not visible.).

The online version of this article includes the following source data and figure supplement(s) for figure 2:

**Source data 1.** Numerical data represented in *Figure 2*.

**Figure supplement 1.** Representative denaturing PAGE data and plots of ln(P/P$_0$) vs. time for the reaction kinetics with an octamer invader, as described in *Figure 2*.

**Figure supplement 2.** Representative denaturing PAGE data and plots of ln(P/P$_0$) vs. time for the reaction kinetics with a hexamer invader as described in *Figure 2*.

below. The reaction rate increased at higher concentrations, but even at 100 µM, $k_{obs}$ was only 0.11 ± 0.03 h$^{-1}$, and we were unable to reach a saturating concentration of hexamer. These results are consistent with the hypothesis that invader binding to the toehold region of the blocker is a necessary first step for strand displacement synthesis, and that saturation of invader/blocker binding leads to the maximum observed rate of strand displacement synthesis. We were therefore curious as to whether lower temperature could promote the reaction at lower invader concentrations and especially for shorter invaders. Lower temperature could facilitate strand displacement synthesis by increasing the stability of the invader/toehold duplex, but could also slow down the rate of the

chemical reaction step, rendering the effects difficult to predict. We therefore conducted the same series of experiments as above, except on ice (blue squares, *Figure 2*). For reactions using the octamer invader, $k_{obs}$ in general decreased by ~50%, reaching a plateau of 0.4 $h^{-1}$. In contrast, when using the hexamer invader, the lower temperature significantly increased the reaction rates, and the plateau now also reached 0.4 $h^{-1}$. These results are consistent with the enhanced binding of a shorter invader to the toehold region at lower temperature facilitating strand displacement synthesis, together with a modest slowing of the overall reaction rate, possibly due to a slower chemical step.

All steps of non-enzymatic strand displacement synthesis are expected to be affected by the concentration of $Mg^{2+}$ in the reaction. $Mg^{2+}$ is thought to catalyze non-enzymatic primer extension in part by deprotonating the 3'-hydroxyl group (*Giurgiu et al., 2018*), and also by promoting the formation of stable RNA duplex structures as well as other effects. We therefore investigated the effect of $Mg^{2+}$ concentration on the rate of primer extension. At room temperature and with a saturating 5 µM concentration of octamer invader, $k_{obs}$ reached a maximum of 2.9 $h^{-1}$ at 400 mM $Mg^{2+}$, compared to 0.5 $h^{-1}$ at 20 mM $Mg^{2+}$ (*Figure 3*). The same trend also occurred for reactions on ice; $k_{obs}$ reached 1.4 $h^{-1}$ with 400 mM $Mg^{2+}$, compared to 0.1 $h^{-1}$ with 20 mM $Mg^{2+}$. High concentrations of $Mg^{2+}$ are known to favor canonical primer extension on open templates, and the effect appears to

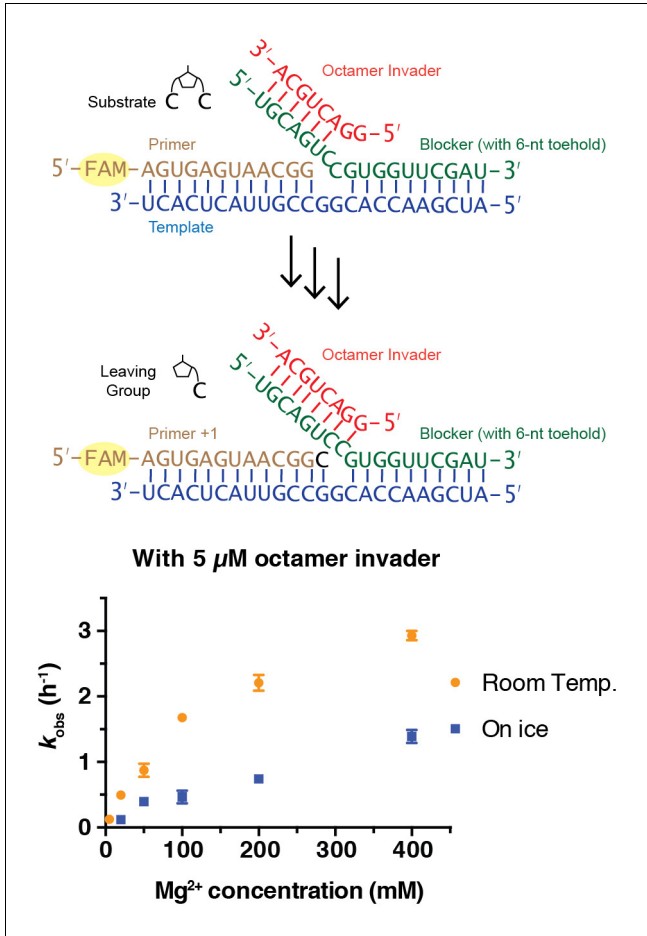

**Figure 3.** Rates of primer extension reactions as a function of $Mg^{2+}$ concentration. All primer extension reactions were conducted in 50 mM $Na^+$-HEPES, pH 8.0, 3 mM C*C, 1.5 µM primer, 2.5 µM template, 3.5 µM blocker and 5 µM octamer invader, at room temperature or on ice as indicated. (n ≥ 3).

The online version of this article includes the following source data and figure supplement(s) for figure 3:

**Source data 1.** Numerical data represented in *Figure 3*.

**Figure supplement 1.** Representative denaturing PAGE data and plots of ln(P/P$_0$) vs. time for the reaction kinetics with an octamer invader, as described in *Figure 3*.

be qualitatively similar for primer extension with strand displacement. The simplest explanation for this observation is that the chemical step of primer extension is enhanced at high concentrations of $Mg^{2+}$, with other effects being relatively minor, at least in the presence of a saturating concentration of invader oligonucleotide. We performed all following experiments, unless otherwise indicated, with 50 mM $Mg^{2+}$, to be consistent with previous work (*Li et al., 2017*).

Following invader-toehold binding and branch migration, the template should be in an open state such that the C*C substrate can bind to the template and react with the primer. However, it is possible that C*C binding to the template could be sterically impaired by the overhanging blocker-invader duplex, and furthermore, once bound to the template the conformation of the C*C could be impacted in a way that might alter its reactivity with the primer. We therefore performed three experiments to disentangle the effects of the toehold and invader on substrate binding and on the rate of the chemical reaction step (*Figure 4*). In the first case (1), the 'blocker' did not contain a toehold region and did not block the GG templating region, only base pairing with the downstream sequence. In the second case (2), the 'blocker' contained an 8 bp toehold region but was not complementary to the GG templating region, due to substitution of AA for CC in the blocker sequence. In this case, the templating region is expected to remain open for substrate binding. The third case

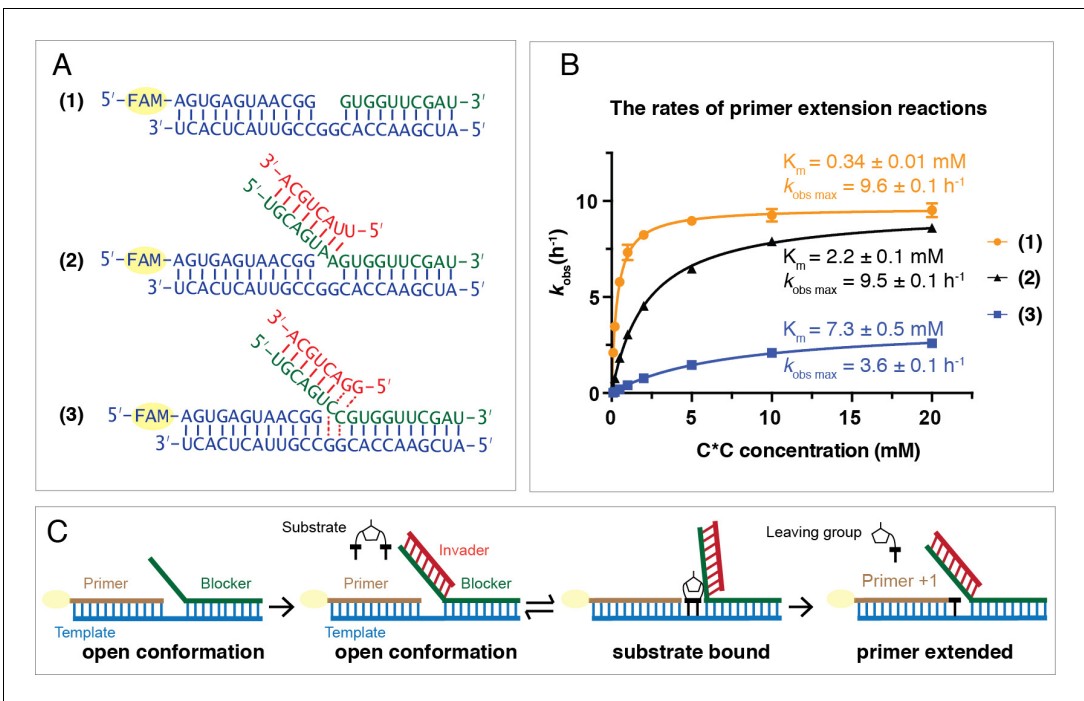

**Figure 4.** C*C dependent primer extension reactions in three different template regimes. (**A**) Three different template configurations. In case (1), the template region GG is open. In case (2), the blocker oligonucleotide cannot base-pair with the template region GG. The complex is constitutively open. In case (3), the template region GG is partitioned between open and closed states. (**B**) Rates of primer extension reactions as a function of C*C concentration. (**C**) An experimental design illustration for case (2). Case (1) and (3) schematics have been shown in *Figure 1B and D*, respectively. All primer extension reactions were conducted at room temperature, 50 mM $Na^+$-HEPES, pH 8.0, 50 mM $MgCl_2$, 1.5 μM primer, 2.5 μM template, 3.5 μM blocker, 0 or 5 μM invader as indicated. Reaction products were analyzed by urea PAGE. (n = 4).

The online version of this article includes the following source data and figure supplement(s) for figure 4:

**Source data 1.** Numerical data represented in *Figure 4*.
**Figure supplement 1.** Representative denaturing PAGE data and plots of ln(P/$P_0$) vs. time for the reaction kinetics in case (1) as described in *Figure 4*.
**Figure supplement 2.** Representative denaturing PAGE data and plots of ln(P/$P_0$) vs. time for the reaction kinetics in case (2) as described in *Figure 4*.
**Figure supplement 3.** Representative denaturing PAGE data and plots of ln(P/$P_0$) vs. time for the reaction kinetics in case (3) as described in *Figure 4*.

(3) represents the strand displacement scenario discussed above with an octamer invader (*Figure 1*). We measured the rate of primer extension as a function of C*C concentration for each of the three scenarios (*Figure 4B*).

For cases (1) and (2), in which the template region is expected to be open, the observed maximum rates ($k_{obs\ max}$) are identical (9.6 ± 0.1 h$^{-1}$ and 9.5 ± 0.1 h$^{-1}$, respectively). This suggests that once the C*C substrate is bound to the GG template region, its reactivity is not affected by the presence or absence of the overhanging invader-blocker duplex. In contrast, when the blocker is able to bind the GG templating region (case 3), $k_{obs\ max}$ drops to 3.6 ± 0.1 h$^{-1}$, ~40% of $k_{obs\ max}$ in (1) and (2). This effect is consistent with a two-state model in which the blocker strand is base-paired to the template GG (and thus preventing C*C binding) about 60% of the time, but is base-paired to the invader GG the other 40% of the time, in which case C*C can bind to the template and once bound reacts normally with the primer. The apparent K$_m$ values for C*C show larger differences across these three regimes. In case (1) where the template is open and there is no overhanging invader-blocker duplex, the K$_m$ for C*C is approximately three times lower (0.34 ± 0.01 mM) than previously observed for binding to a primer template complex with no blocker strand at all (*Walton and Szostak, 2017*). This suggests that the binding of C*C is stabilized by stacking with the downstream blocker strand. In contrast, the presence of an overhanging invader-blocker duplex increases the K$_m$ for C*C roughly 6-fold to 2.2 ± 0.1 mM, suggesting that the presence of an invader-blocker duplex sterically interferes with C*C binding. Finally, for case (3), in which the blocker is partitioned between base-pairing with the template or the invader, the binding of C*C to the template is further impaired and the apparent K$_m$ is 7.3 ± 0.5 mM. The reason for this further increase in K$_m$ is not obvious, but could reflect subtle changes in the extent of steric blocking due to the altered sequence context near the bound C*C, or perhaps more complex effects not fully accounted for by a simple two state model for blocker partitioning.

Having developed a basic understanding of single-step strand displacement synthesis, we asked whether it is possible to achieve multiple steps of primer extension using multiple invaders. For the addition of multiple nucleotides, the invader must be able to dissociate from the blocker after the formation of the primer +1 product, to allow the second invader to base pair with the blocker and open the next templating region so that the +2 product can be made, and so forth for multiple reaction cycles. For these experiments we decided to adopt hexamer invaders. Although an octamer invader results in a higher $k_{obs}$ than a hexamer invader at room temperature, the 8 nt invader/blocker duplex may be too stable to dissociate rapidly at room temperature. We reasoned that the faster on and off rate of a hexamer invader could be beneficial in the context of multiple addition reactions. To partially compensate for the weaker binding of hexamer invaders, we used 400 mM Mg$^{2+}$ instead of 50 mM Mg$^{2+}$ and replaced U with 2sU for these experiments, because 2sU forms a stronger base-pair with A than U (*Heuberger et al., 2015*). We synthesized seven different hexamer invaders (*Figure 5A*) to allow for seven steps of strand displacement synthesis. Instead of preparing all possible imidazolium-bridged-dinucleotides, we combined C*C with 2-amino-imidazole activated guanosine (2-AIpG) and 2-amimo-imidazole activated 2-thio-uridine (2-AIp2sU), which rapidly equilibrate within the reaction mixture to form all required imidazolium-bridged dinucleotides. Without a bound blocker, RNA primer extension was poor, although several faint bands above +1 products were present (*Figure 5B*). This is as expected, because the efficient copying of mixed sequence RNA templates requires activated short oligonucleotide helpers (*Li et al., 2017*; *Prywes et al., 2016*). In the presence of the blocker without invaders, no primer extension was observed, as expected, because the imidazolium-bridged-dinucleotides cannot bind to the occluded template. With blocker and seven hexamer invaders, we observed unambiguous +1 and +2 primer extension products, and several faint bands above as well. This result indicates that multiple nucleotide additions are indeed possible during primer extension with strand displacement. The observation that primer extension is improved in the strand displacement context *versus* a completely open template suggests that the binding and/or reactivity of the imidazolium-bridged-dinucleotides may be enhanced when they are sandwiched in between the primer and the blocker strand.

Because the slow rate of non-enzymatic RNA copying on mixed-sequence templates made it difficult to clearly demonstrate continued iterative steps of strand displacement synthesis, we repeated the above experiments with 2-aminoimidazole activated 3′-amino-2′,3′-dideoxyribo-nucleotide monomers (3′-NH$_2$-2AIpddN). We have recently shown that these monomers polymerize rapidly on an RNA template, forming a 3′-NP-DNA complementary strand (*O'Flaherty et al., 2019*;

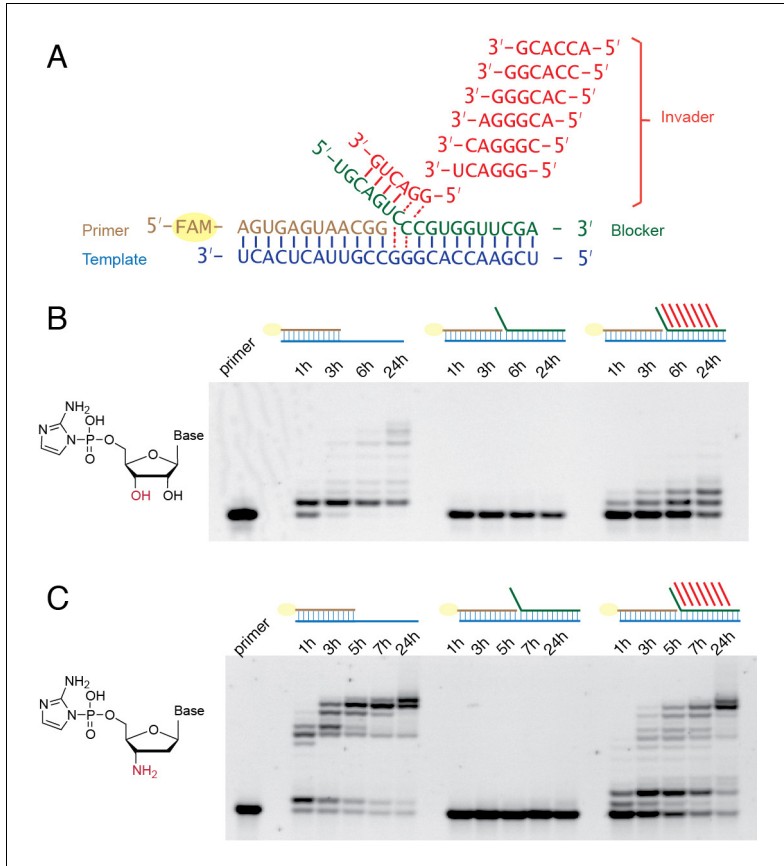

**Figure 5.** Multi-step primer extension with strand displacement. (**A**) Illustration of the experimental configuration with seven different hexamer invaders. (**B**) PAGE analysis of primer extension with RNA monomers. The reactions were conducted at room temperature, 50 mM Na⁺-HEPES, pH 8.0, 400 mM MgCl₂, 1.5 µM primer, 2.5 µM template, with or without 3.5 µM blocker as indicated, with or without 20 µM each hexamer invader as indicated, 5 mM C*C, 10 mM 2-AIpG and 10 mM 2-AIp2sU. (**C**) PAGE analysis of primer extension with 3′-amino-2′,3′-dideoxyribonucleotide monomers. Reactions were conducted as in (**B**), except that 10 mM 3′-NH₂-2AIpddA, 10 mM 3′-NH₂-2AIpddG and 10 mM 3′-NH₂-2AIpddT were used instead of RNA monomers, and the 3′-amine primer was used instead of the RNA primer.

*Zhang et al., 2013a*; *Zhang et al., 2013b*). We also used a primer in which the last nucleotide was also substituted with a 3′-NH₂-2′,3′-dideoxyribo-nucleotide. In the reaction without any blocker or invaders, the full-length product appeared within 5 hr. In the presence of the blocker but absence of invaders, no primer-extension occurred. In the presence of both the blocker and the seven hexamer invaders, the majority of the primer was extended to the full-length product within 24 hr (*Figure 5C*).

To investigate whether the multiple addition products observed in these experiments resulted from primer extension with strand displacement, or because the blocker had failed to form a stable duplex with the template under these conditions, we adapted a fluorescence-quencher assay (*Larsen et al., 2016*). In this assay the primer was not fluorescently labeled, but the template was labeled with Cyanine 3 at its 5′-end and the blocker was modified with Black Hole Quencher−2 at its 3′-end. When the blocker forms a duplex with the template, the fluorescence of Cy3 is quenched. When the blocker dissociates from the template, the fluorescence is recovered (*Figure 6A*). When we added an unlabeled RNA strand that was complementary to the template prior to addition of the labeled blocker, binding of the blocker was inhibited, and fluorescence remained the same high level throughout the time course of 48 hr (*Figure 6B*). In three negative control reactions, 3′-NH₂-2AIpddN monomers and/or invaders were omitted, and we observed very low fluorescent signal throughout the reaction time courses, indicating that the blocker had remained base-paired with the

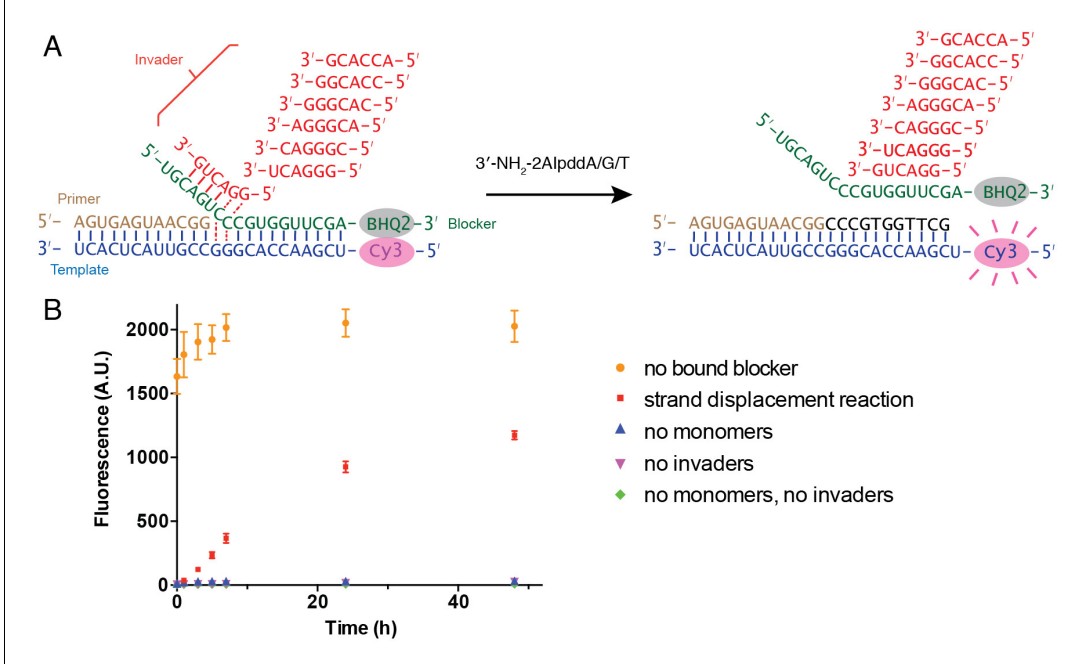

**Figure 6.** Fluorescence-quencher assay for non-enzymatic primer extension with strand displacement. (**A**) Illustration of the experimental design. The template was labeled with Cy3 at its 5'-end. The blocker was modified by Black Hole Quencher—2 at the 3'-end. When the blocker forms a stable duplex with the template, the fluorescent signal of Cy3 is quenched. In the presence of invaders and 3'-NH$_2$-2AIpddN, the primer extends, resulting in blocker dissociation and increasing Cy3 signal. (**B**) Time course of the fluorescent signal in five different experiment groups. Strand displacement reactions contained 1.5 μM template, 2 μM blocker, 2.5 μM primer, 50 mM Na$^+$-HEPES, pH 8.0, 50 mM MgCl$_2$, 20 μM each hexamer invader and 10 mM 3'-NH$_2$-2AIpddA/G/T. Positive control reactions contained the same components, with the addition of 2 μM of complementary strand RNA to the template, so that the blocker was unbound. The negative control reactions contained the same components but omitted monomers and/or invaders, as indicated. The experiments were performed at room temperature in triplicate.

The online version of this article includes the following source data for figure 6:

**Source data 1.** Numerical data represented in *Figure 6*.

template. Finally, in a complete reaction mixture containing all components, the fluorescence signal grew to ~50% of that observed in the positive control in 24 hr, and ~60% in 48 hr. These results confirm that a strand displacement mechanism is operating and is required for successful primer extension under these conditions.

## Discussion

Our experiments show that non-enzymatic primer extension by strand displacement synthesis can be facilitated by short RNA oligonucleotides, resulting in the template-directed synthesis of RNA (and the related polymer 3'-NP-DNA) without prior strand separation. The idea of using strand displacement synthesis for replication provides an alternative to the more commonly considered model in which the copying of a template leads to a duplex whose strands must be separated and remain separated in order to be copied again. Strand separation models are plagued by the mismatch between the comparatively slow rate of non-enzymatic template copying and the much faster kinetics of strand reannealing (except at extremely low strand concentrations). In contrast, strand displacement synthesis should proceed well at high strand concentrations, allowing both genomic and functional strands to build up to high enough levels to have strong phenotypic influences on the host protocell. Replication by strand displacement synthesis is also appealing in that it is closer to the mode of replication universally employed in biology, and because it should operate in a complex prebiotic context of concentrated and heterogeneous nucleotides and oligonucleotides.

The mechanism of strand displacement synthesis that we have explored involves short oligonucleotides, which act as 'invaders' to open up the template region to be copied through a toehold/branch migration process. We propose two prebiotically relevant sources for such oligonucleotides. First, the hydrolysis of longer oligonucleotides could generate multiple short oligonucleotides that can serve as invaders. Second, incomplete or partial products of template copying could dissociate from the template and act as invaders. Such partial replication products would include stalled primer extension products, resulting from the incorporation of mismatched monomers or non-canonical nucleotides (*Kim et al., 2018*; *Rajamani et al., 2010*). Additionally, other prebiotically plausible genetic polymers (TNA, ANA, DNA, etc.) and heterogeneous oligonucleotides that could have coexisted alongside oligoribonucleotides could also promote strand displacement via the formation of invaders. Recent work in our lab (*Kim et al., 2019*) has shown that oligonucleotides ending in a 3'-arabino nucleotide cannot be further extended, providing a mechanism for the generation of short invaders that are inert to copying chemistry. Thus, the products of both degradative reactions and stalled replicative processes could contribute an essential function to genomic replication. In the context of a protocell, the set of such small oligonucleotides would not include all possible sequences, but would derive from and therefore be biased towards the genomic sequences of the host protocell. We have previously suggested that the accumulation of such fragments within a primitive protocell could serve useful homeostatic regulatory functions, by acting as reversible concentration dependent inhibitors or activators of ribozymes (*Engelhart et al., 2016*).

We propose that non-enzymatic strand displacement synthesis could potentially enable or at least contribute to complete cycles of RNA replication. However, three major problems must be overcome before complete replication cycles can be demonstrated in the laboratory. First is the problem of initiation. In the presence of a defined primer, a transient thermal fluctuation might allow a primer to bind to a template and begin strand displacement synthesis, but how initiation might begin in the absence of defined primers remains an open question. Second, cycles of replication would require the copying of both complementary strands, which would seem to require the presence of invaders derived from both strands. Whether strand displacement synthesis could proceed in the presence of invaders derived from complementary strands must be examined experimentally. We suggest that frequent thermal fluctuations might allow invader-invader and invader-blocker duplexes to dissociate and then re-form, allowing for one to a few nucleotides of primer extension with each cycle of reorganization. Finally, the slow rate of non-enzymatic primer extension with ribonucleotides is a problem for all models of non-enzymatic RNA replication. As a proxy for RNA and ribonucleotides, the more reactive 3'-NP-DNA and 3'-aminonucleotides have enabled us to demonstrate multi-step strand displacement synthesis. However, an important goal for laboratory studies is to make template directed RNA synthesis as efficient as 3'-NP-DNA synthesis. One possibility under investigation in our lab is that prebiotically relevant small molecules or short peptides could act as catalysts for the copying reaction by promoting the binding or positioning of the catalytic metal ion within the reaction center. However, we note that in a strand displacement synthesis model, genomic RNA fragments remain largely double stranded and thus protected from degradation, which could make replicative synthesis possible even if primer extension is quite slow.

Beyond the scope of replication, we propose that strand displacement could potentially be involved in the regulated synthesis of primitive ribozymes. After the first round of copying, any ribozyme sequence would remain sequestered as a stable duplex with its daughter strand and would be unable to perform its catalytic function. However, primer extension with strand displacement could liberate the ribozyme from its complementary strand, enabling it to fold into a tertiary structure and execute its catalytic function. Hence strand displacement reactions with invader oligonucleotides that can rapidly associate and dissociate could allow both the rate of RNA replication and ribozyme catalytic activity to be controlled.

In summary, our work demonstrates the concept of non-enzymatic RNA copying with strand displacement, offering a novel approach to RNA self-replication that relies on the presence of RNA fragments derived from degradation or partial template copying. If the problems of initiation and replication of both strands could be overcome, it might be possible to initiate the non-enzymatic evolution of RNA inside model protocells, mimicking the origin of life.

# Materials and methods

## Oligonucleotides synthesis and purification

Chemical reagents were purchased from Chemgenes (Wilmington, MA) and Glen Research (Sterling, VA). Oligonucleotides were prepared by solid-phase synthesis on an Expedite 8909 or ABI 394 DNA/RNA synthesizer. RNAs were deprotected by standard methods. Non-dye labeled oligonucleotides were purified by GlenPak columns. Dye-labeled oligos were purified by polyacrylamide gel electrophoresis and desalted on Sep-Pak C18 cartridges from Waters (Milford, MA). Oligonucleotides were analyzed by high resolution mass spectrometry (HRMS) on an Agilent 6520 QTOF LC-MS in house.

## C*C synthesis and purification

0.3 mmol CMP (Sigma) was mixed with 0.15 mmol 2-amino-imidazole•HCl (Combi-blocks) in 5 ml anhydrous DMSO (Sigma) and 0.4 ml TEA (Sigma). Then 1 g triphenylphosphine (Sigma) and 0.88 g 2,2'-dipyridyldisulfide (Combi-blocks) were added in order and stirred vigorously. The reaction was continued in a sealed container for 20 min. The product C*C was precipitated by adding 40 ml acetone and 2 ml $NaClO_4$-saturated acetone. After centrifugation at 3500 rpm for 10 min, the pellet was washed with 40 ml acetone:diethyl ether (1:1) and centrifuged again. The pellet was washed twice, then dried under house vacuum to remove organic solvent. The dry pellet was resuspended in 20 mM TEAB pH 8.0 and purified on a 50 g C18Aq column over 12 CV of 0–10% acetonitrile in 2 mM TEAB buffer (pH 8.0). The product was analyzed by $^{31}$P-NMR and low resolution mass spectrometry (LRMS) before being aliquoted and lyophilized.

## 2-AIpG synthesis and purification

2-AIpG was prepared by the same procedures as C*C, except that 0.3 mmol GMP (Sigma) and 3 mmol2-amino-imidazole•HCl were used in the reaction. The dry pellet was resuspended in 20 mM TEAB pH 8.0 and purified on a 50 g C18Aq column over 12 CV of 0–10% acetonitrile in 2 mM TEAB buffer (pH 8.0). After purification, the 2-AIpG solution was adjusted to pH ~10 with NaOH before being aliquoted and lyophilized.

## 2-AIp2sU synthesis and purification

2-AIp2sU was prepared according to previously published procedures (*Li et al., 2017*), as summarized below.

**Chemical structure 1.** 2-AIp2sU synthesis.

The purification procedure is the same as 2-AIpG.

## 3'-NH$_2$-2AIpddA/G/T synthesis and purification

3'-NH$_2$-2AIpddA, G, and T were prepared according to previously published procedures (*O'Flaherty et al., 2019*), as summarized below.

**Chemical structure 2.** 3'-NH$_2$-2AIpddA/G/T synthesis.

The purification procedure is the same as 2-AIpG.

## Primer extension reaction and PAGE analysis

The primer-template and primer-template-blocker complexes were prepared in a solution containing 7.5 µM primer, 12.5 µM template, 0 or 17.5 µM blocker, 50 mM Na$^+$-HEPES (pH 8.0), 50 mM NaCl and 1 mM EDTA (pH 8.0) by heating at 95℃ for 30 s and slowly cooling down to 25℃. The annealed product was diluted five fold in primer extension reaction. Stock solutions of C*C and monomers used in the reaction were prepared freshly and adjusted to pH 8.0 immediately before the reaction. At each time point, 0.5 µl of reaction sample was added to 25 µl quenching buffer, containing 8M Urea, 20 mM EDTA, 1x TBE and 10 µM complementary RNA of template. Primer extension products were resolved by 20% (19:1) denaturing PAGE with 7 M urea. The gel was scanned using a Typhoon 9410 scanner, and the bands were quantified using the ImageQuant TL software.

## Fluorescence-quencher assay

All components were mixed in 96 well, half-area, black, polystyrene plates (Costar). Fluorescence signals were recorded at an excitation wavelength of 535 nm and an emission wavelength of 595 nm using a SpectraMax i3 plate-reader.

## Acknowledgements

JWS is an Investigator of the Howard Hughes Medical Institute. This work was supported in part by a grant (290363) from the Simons Foundation to JWS and by a grant from the NSF (CHE-1607034) to JWS. DKO is a recipient of a Postdoctoral Research Scholarship (B3) from the Fonds de Recherche du Québec−Nature et Technologies (FRQNT), Québec, Canada, and a Postdoctoral Fellowship from Canadian Institutes of Health Research (CIHR) from Canada. The authors thank Dr. Li Li, Dr. Victor S Lelyveld, Dr. Fanny Ng, Dr. Andrew J Bendelsmith and Dr. Russell Algera for helpful discussion and technical assistance. The authors also thank the Szostak group for helpful feedback.

## Additional information

### Funding

| Funder | Grant reference number | Author |
| --- | --- | --- |
| Simons Foundation | 290363 | Jack W Szostak |
| National Science Foundation | CHE-1607034 | Jack W Szostak |
| Fonds de Recherche du Québec - Nature et Technologies | Postdoctoral Research Scholarship (B3) | Derek K O'Flaherty |
| Canadian Institutes of Health Research | Postdoctoral Fellowship | Derek K O'Flaherty |

The funders had no role in study design, data collection and interpretation, or the decision to submit the work for publication.

### Author contributions

Lijun Zhou, Conceptualization, Investigation, Visualization, Writing—original draft, Writing—review and editing; Seohyun Chris Kim, Katherine H Ho, Derek K O'Flaherty, Constantin Giurgiu, Tom H Wright, Investigation, Writing—review and editing; Jack W Szostak, Conceptualization, Resources, Supervision, Funding acquisition, Writing—review and editing

### Author ORCIDs

Lijun Zhou https://orcid.org/0000-0002-0393-4787
Seohyun Chris Kim https://orcid.org/0000-0002-2230-1774
Derek K O'Flaherty https://orcid.org/0000-0003-3693-6380
Constantin Giurgiu https://orcid.org/0000-0003-0145-0110
Tom H Wright https://orcid.org/0000-0003-2231-8223
Jack W Szostak https://orcid.org/0000-0003-4131-1203

**Decision letter and Author response**

Decision letter https://doi.org/10.7554/eLife.51888.sa1

Author response https://doi.org/10.7554/eLife.51888.sa2

## Additional files

### Supplementary files

• Transparent reporting form

### Data availability

All data generated or analysed during this study are included in the manuscript and supporting files.

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
