## [Decision Letter]

**Acceptance summary:**

It is widely accepted that RNA was the informational macromolecule necessary for the origin of life. Nevertheless, the "RNA hypothesis" has several hurdles to overcome including the replication problem. Because copying of any nucleic acid polymer results in a double stranded molecule, it has been difficult to envision a mechanism that would replicate the dsRNA. In this paper Szostak and colleagues use conditions that promote "replication". Accordingly, this work represents a substantive advance in origin of life research.

**Decision letter after peer review:**

Thank you for submitting your article "Non-enzymatic primer extension with strand displacement" for consideration by *eLife*. Your article has been reviewed by three peer reviewers, including Timothy W Nilsen as the Reviewing Editor and Reviewer #1, and the evaluation has been overseen by Michael Marletta as the Senior Editor. The following individual involved in review of your submission has agreed to reveal their identity: John C Chaput (Reviewer #2).

The reviewers have discussed the reviews with one another and the Reviewing Editor has drafted this decision to help you prepare a revised submission.

All three reviewers found the work to be a significant and substantive step forward in origin-of-life research. As such, the manuscript is in principle appropriate for publication in *eLife*. Nevertheless, all of the reviewers, particularly reviewers 1 and 3, have made a number of suggestions to improve the accessibility and scientific accuracy of the presentation. Please address these points as thoroughly as possible.

Reviewer #1:

Here, Szostak and colleagues provide compelling evidence that under certain conditions they can observe copying of a template RNA that is sequestered in an RNA-RNA duplex. Copying is via primer extension. These findings represent a substantive step forward in the "replication problem" of prebiotic nucleic acids.

While the data are unambiguous, I have several comments that if addressed would improve the readability of this manuscript by a broad audience.

1) I found the diagrams to be inadequate. A large schematic of the constructs under study would be quite helpful. Such diagrams should explicitly label the invader, blocker, toehold, etc., to provide a clear picture of the experimental design. There should be a series of schematics first illustrating invasion and culminating with addition of the C residue.

2) Similarly detailed schematics should accompany Figures 2, 3, 4, 5 and 6 such that it becomes clear what is different in each figure. The legends are quite brief and not very informative.

3) The term catalysis is used loosely throughout the paper. Just as one example, the authors state "can be catalyzed by short RNA oligonucleotides". I do not see how oligonucleotides per se can be viewed as catalysts. The authors should examine the use of the word throughout the paper.

Reviewer #2:

This manuscript provides an interesting solution to the strand displacement problem, which is a major obstacle facing origins of life researchers that are attempting to replicate RNA in the absence of biological enzymes. The crux of the problem lies in the fact that the product of a non-enzymatic primer-extension reaction is a double-stranded (ds) RNA molecule whose strands must be separated in order to complete the replication cycle. Although dsRNA can be denatured with heat, the rate of reannealing is much fast than the rate of template copying. Thus, the process of non-enzymatic replication is product inhibited. Here, Zhou et al. describe a prebiotically plausible solution to the strand displacement problem that involves a branch migration mechanism in which short 'invader' strands are used to unwind one of the RNA strands so that non-enzymatic primer-extension can continue. The authors first explore the concept of branch migration in the context of single-nucleotide addition reactions. They then proceed to more challenging examples, including a mixed-template example that involves the extension of multiple different nucleotides. Undoubtedly, this work represents a significant step forward in the realization of abiotic RNA replication.

Reviewer #3:

The study addresses a central problem in RNA World research, which is the need for strand displacement in the replication of an RNA sequence. The study shows a careful analysis of the possibility that 'invader oligonucleotides' could allow strand displacement during primer extension. It first focuses on this process with single nucleotide primer extension, then looks at the mechanism, then at multiple nucleotides extension. The text is well-written in the Results section but needs to be improved in the Abstract and Introduction.

1) There are quite a few instances where the wording is incorrect or needs to be toned down, especially in Abstract and Introduction.

2) The impact of the paper would be increased if the proposal made in the Discussion – primer extension by strand displacement in the presence of invader oligos for both template and primer strand (Discussion, third paragraph) would be shown experimentally.

---

## [Author Response]

Reviewer #1:[…] While the data are unambiguous, I have several comments that if addressed would improve the readability of this manuscript by a broad audience.1) I found the diagrams to be inadequate. A large schematic of the constructs under study would be quite helpful. Such diagrams should explicitly label the invader, blocker, toehold, etc., to provide a clear picture of the experimental design. There should be a series of schematics first illustrating invasion and culminating with addition of the C residue.2) Similarly detailed schematics should accompany Figures 2, 3, 4, 5 and 6 such that it becomes clear what is different in each figure. The legends are quite brief and not very informative.

We have modified Figures 1, 2, 3 and 4 by adding expanded diagrams with clear labeling of all components. We have added labels to Figures 5 and 6, but otherwise left them unchanged. To avoid excessive repetition, we would prefer not to include all of the details of each step in each figure, focusing on the experimental design and results.

3) The term catalysis is used loosely throughout the paper. Just as one example, the authors state "can be catalyzed by short RNA oligonucleotides". I do not see how oligonucleotides per se can be viewed as catalysts. The authors should examine the use of the word throughout the paper.

The oligonucleotide invaders are catalysts in that they accelerate a chemical reaction while remaining unchanged themselves. We believe our use of the word was appropriate, as these short RNA oligonucleotides lower the energetic barrier to strand displacement synthesis and can potentially engage in multiple turnover synthesis. However, since they are not acting directly on the chemical step, but only on conformational changes, to avoid confusion we have avoided the use of the terms catalyst and catalyzed as suggested. We have therefore edited the manuscript as follows:

Abstract: “catalysed” replaced with “facilitated”

Results, first paragraph: “uncatalyzed” replaced with “spontaneous”

Results, first paragraph: “catalysis” replaced with “an auxiliary’

Results, second paragraph: “catalyzed” replaced with “facilitated”

Discussion, first paragraph: “catalyzed” replaced with “facilitated”

Reviewer #3:[…] The text is well-written in the Results section but needs to be improved in the Abstract and Introduction.1) There are quite a few instances where the wording is incorrect or needs to be toned down, especially in Abstract and Introduction.

We have revised our manuscript as suggested.

2) The impact of the paper would be increased if the proposal made in the Discussion – primer extension by strand displacement in the presence of invader oligos for both template and primer strand (Discussion, third paragraph) would be shown experimentally.

Although we agree that this would be highly significant, it is also much more complex than might appear at first glance. The problem is that complementary invaders will associate to form duplexes, lowering the concentration of free invader strands. It may be possible to minimize this effect by optimizing salt and temperature conditions, or even to avoid this problem altogether through the use of invader strands containing deoxynucleotides or other variant nucleotides known to lower duplex melting temperatures. Alternatively, invader-catalyzed strand displacement may only be needed for a short stretch of template copying, followed by dissociation of the rest of the complementary blocker strand; in this case, invader strands might only be needed for short non-overlapping stretches of the duplex template. Sorting out these and other possibilities will involve extensive additional work, which will be the subject of future publications. We therefore consider this type of experiment to be beyond the scope of the present study.